# The Virome of Acute Respiratory Diseases in Individuals at Risk of Zoonotic Infections

**DOI:** 10.3390/v12090960

**Published:** 2020-08-29

**Authors:** Nguyen Thi Kha Tu, Nguyen Thi Thu Hong, Nguyen Thi Han Ny, Tran My Phuc, Pham Thi Thanh Tam, H. Rogier van Doorn, Ho Dang Trung Nghia, Dang Thao Huong, Duong An Han, Luu Thi Thu Ha, Xutao Deng, Guy Thwaites, Eric Delwart, Anna-Maija K. Virtala, Olli Vapalahti, Stephen Baker, Le Van Tan

**Affiliations:** 1Doctoral School in Health Sciences, Faculty of Medicine, University of Helsinki, 00014 Helsinki, Finland; olli.vapalahti@helsinki.fi; 2Oxford University Clinical Research Unit, Ho Chi Minh City 7000, Vietnam; hongntt@oucru.org (N.T.T.H.); nynth@oucru.org (N.T.H.N.); phuctm@oucru.org (T.M.P.); tamptt@oucru.org (P.T.T.T.); nghiahdt@oucru.org (H.D.T.N.); huongdt@oucru.org (D.T.H.); gthwaites@oucru.org (G.T.); 3Dong Thap Provincial Center for Disease Control, Cao Lanh City 660273, Dong Thap Province, Vietnam; anhanduong@gmail.com (D.A.H.); luuthithuha2018@gmail.com (L.T.T.H.); 4Oxford University Clinical Research Unit, Ha Noi 8000, Vietnam; rvandoorn@oucru.org; 5Centre for Tropical Medicine and Global Health, Nuffield Department of Medicine, University of Oxford, Oxford OX3 7LG, UK; 6Department of Laboratory Medicine, University of California, San Francisco, CA 94143, USA; XDeng@bloodsystems.org (X.D.); Eric.Delwart@ucsf.edu (E.D.); 7Vitalant Research Institute, San Francisco, CA 94118, USA; 8Department of Veterinary Biosciences, Faculty of Veterinary Medicine, University of Helsinki, 00014 Helsinki, Finland; anna-maija.virtala@helsinki.fi; 9Virology and Immunology, HUSLAB, Helsinki University Hospital, 00029 Helsinki, Finland; 10Cambridge Institute of Therapeutic Immunology & Infectious Disease (CITIID), Department of Medicine, University of Cambridge, Cambridge CB2 0QQ, UK; sgb47@medschl.cam.ac.uk

**Keywords:** virome, acute respiratory disease, NGS, metagenomics, zoonoses, novel cyclovirus, novel statovirus, novel gemycircularvirus

## Abstract

The ongoing coronavirus disease 2019 (COVID-19) pandemic emphasizes the need to actively study the virome of unexplained respiratory diseases. We performed viral metagenomic next-generation sequencing (mNGS) analysis of 91 nasal-throat swabs from individuals working with animals and with acute respiratory diseases. Fifteen virus RT-PCR-positive samples were included as controls, while the other 76 samples were RT-PCR negative for a wide panel of respiratory pathogens. Eukaryotic viruses detected by mNGS were then screened by PCR (using primers based on mNGS-derived contigs) in all samples to compare viral detection by mNGS versus PCR and assess the utility of mNGS in routine diagnostics. mNGS identified expected human rhinoviruses, enteroviruses, influenza A virus, coronavirus OC43, and respiratory syncytial virus (RSV) A in 13 of 15 (86.7%) positive control samples. Additionally, rotavirus, torque teno virus, human papillomavirus, human betaherpesvirus 7, cyclovirus, vientovirus, gemycircularvirus, and statovirus were identified through mNGS. Notably, complete genomes of novel cyclovirus, gemycircularvirus, and statovirus were genetically characterized. Using PCR screening, the novel cyclovirus was additionally detected in 5 and the novel gemycircularvirus in 12 of the remaining samples included for mNGS analysis. Our studies therefore provide pioneering data of the virome of acute-respiratory diseases from individuals at risk of zoonotic infections. The mNGS protocol/pipeline applied here is sensitive for the detection of a variety of viruses, including novel ones. More frequent detections of the novel viruses by PCR than by mNGS on the same samples suggests that PCR remains the most sensitive diagnostic test for viruses whose genomes are known. The detection of novel viruses expands our understanding of the respiratory virome of animal-exposed humans and warrant further studies.

## 1. Introduction

Acute respiratory infections are the leading cause of morbidity and mortality worldwide [1,2]. Despite intensive laboratory investigations, a substantial proportion of acute respiratory infections are of unknown etiology, resulting in difficulties in clinical management [3,4,5,6]. Metagenomics is an unbiased (independent of specific sequences) approach increasingly being applied in virus discovery as well as for molecular diagnostics [6,7,8]. 

Viruses are the main causes of acute respiratory infections with the potential to cause pandemics [9,10,11,12]. Notably, most emerging viral agents of acute respiratory diseases are of zoonotic origin and pose a major threat to human health [10,11,12,13]. The ongoing coronavirus disease 2019 (COVID-19) pandemic caused by severe acute respiratory syndrome coronavirus 2 (SARS-CoV-2) exemplifies problems resulting from emerging zoonotic pathogens [9]. Individuals with frequent and sustained contact with animals are considered at higher risks of infections with zoonotic pathogens and are therefore suitable targets for emerging virus surveillance programs [13]. The early detection of emerging viral pathogens of animal origin that exhibit potential for human-to-human transmission remains a difficult but essential step to mitigate their propagation. 

Here, we characterized the eukaryotic virome of respiratory specimens taken from patients presenting with acute respiratory infections in a cohort with a high level of animal exposure [14] in southern Vietnam. Additionally, we compared viral detection by mNGS versus PCR to assess the utility of mNGS in routine diagnostics.

## 2. Materials and Methods

### 2.1. Ethical Approvals

All study subjects gave their informed consent for inclusion before participating in the study. The study was approved by the Oxford Tropical Research Ethics Committee (OxTREC) (No. 157-12), the United Kingdom, and the Ethic Committees of Dong Thap Hospital in Dong Thap provinces, Vietnam.

### 2.2. The Sentinel Cohort Study and Samples

The clinical samples used in this study were derived from a sentinel cohort study described previously [5,14]. The cohort study is a community-based component of The Vietnamese Initiative on Zoonotic Infections (VIZIONS) project [15,16], which was conducted to detect potential zoonotic transmission. In brief, individuals, including animal-raising farmers, slaughterers, animal health workers, and rat-traders from Dong Thap and Dak Lak provinces in Vietnam, were recruited in the cohort study and followed for 3 years, 2013 to 2016 [14].

Starting each study year, to create baseline data, the cohort members were interviewed and them plus their animals (all without symptoms and sign of respiratory disease) were sampled. During the follow-up period, whenever the cohort member got any signs/symptoms of respiratory tract infections and fever (≥38 °C), specimens from the diseased individual and their animals were collected. The clinical specimens collected from each cohort member and their animals consisted of rectal, pooled nasal and throat swabs, and blood [5,14]. Here, we focused on nasal-throat swabs sampled at respiratory disease episodes during 2013 (Figure 1).

### 2.3. Metagenomic Next-Generation Sequencing (MNGs) Assay

Initially, 200 µ of nasal-throat swabs collected at disease episodes and a negative control containing viral transport medium were first treated with 20 U of turbo DNase (Ambion, Life Technology, Carlsbad, CA, USA) and 50 U of RNase I (Ambion) at 37 °C for 30 min [17]. Viral RNA was then isolated from nuclease-treated materials using a QIAamp 96 Virus QIAcube HT Kit (QIAGEN GmbH, Hilden, Germany), following the manufacturer’s instructions for nucleic acid extraction. The nucleic acid output was then recovered in 50 μL of elution buffer (provided with the QIAamp kit).

Double-stranded DNA synthesis, random amplification, and library preparation were carried out as previously described [6]. The prepared library was sequenced using the MiSeq reagent kit V3 in an Illumina MiSeq platform (Illumina, San Diego, CA, USA. The double indexes of Nextera XT Index Kit (Illumina) was used to multiplex and differentiate the samples in each run.

### 2.4. Analysis of mNGS Sequence Data

An in-house analysis pipeline was used to analyze sequence data that is posted in GitHub: https://github.com/xutaodeng/virushunter/. Briefly, the adaptors, low-quality reads, and duplicate reads were firstly removed. The reads related to human and bacterial genomes were subtracted by mapping reads using bowtie2 (version 2.2.4) to concatenated human reference genome sequence and mRNA sequences (hg38), and bacterial nucleotide sequences extracted from NCBI nt fasta file (ftp://ftp.ncbi.nlm.nih.gov/blast/db/FASTA/, February 2019) based on NCBI taxonomy (ftp://ftp.ncbi.nih.gov/pub/taxonomy, February 2019) [18]. The remaining reads were de novo assembled using ENSEMBLE software [19], which uses a partitioned subassembly approach to integrate the use of various de Bruijn graph (DBG) and overlap-layout-consensus assemblers (OLC) [19]. To allow for sensitive screening of viral sequences, the resulting contigs (plus single reads) were aligned against the viral proteome of the NCBI’s RefSeq and the viral proteome of the non-redundant database by the Basic Local Alignment Search Tool (BLASTx). Matches with E score <0.01 were retained. To filter out tentative viral hits that showed better alignments to non-viral sequences, these tentative matches to viral proteins were then aligned to the GenBank’s entire non-redundant proteome database using DIAMOND algorithm version 0.9.6 [20]. Sequences were then classified as viral or removed as non-viral according to the NCBI taxonomy of the best hits (lowest E score) in the non-redundant proteome database. Viral reads described here have E scores to viral proteins <10^−10^. 

### 2.5. PCR Confirmatory Testing of Viruses Detected by Metagenomics and Genome Sequencing

mNGS-detected viruses that were previously reported in human samples were confirmed by specific RT-PCR using previously published or newly designed primers/probes, followed by Sanger sequencing of the obtained amplicons (if applicable). The PCR confirmatory experiments were carried out on newly extracted nucleic acid from original patient samples using MagNApure 96 platform (Roche Diagnostics, Mannheim, Germany) [5]. Inverse primers were used to amplify and then sequence complete circular virus genomes.

### 2.6. PCR Screening by New Primers Designed Based on mNGS Contigs

For eukaryotic viral genomes detected in mNGS output, PCRs (confirmed by Sanger sequencing) were applied to screen for their genetic sequences. The primers were designed based on the mNGS contigs of the viruses of interest and PCRs conducted on mNGS-negative samples (Figure 1). PCR screening was carried out on the nucleic acid re-isolated from the original respiratory sample using a MagNApure 96 platform (Roche Diagnostics) [5].

### 2.7. Viral Genotyping

Sequences related to the enterovirus genome were classified using the Enterovirus Genotyping Tool Version 1.0 [21]. For influenza virus A, coronavirus (CoV), and respiratory syncytial virus (RSV)-A, the read sequences for subtyping [22,23,24] were located and extracted from mNGS output using the Map-to-reference tool of Geneious Prime 2020.0.2 software (Biomatters, Auckland, New Zealand). The recovered sequences were then used to compare against the NCBI non-redundant protein sequence database using BLASTx (*E* value ≤10^−5^). 

### 2.8. Phylogenetic Analysis

Sequence alignment was conducted by the MUSCLE algorithm of MEGA software version X. Phylogenetic trees were built by the Maximum Likelihood (bootstrap 1000) algorithm of the MEGA.

### 2.9. Nucleotide Sequence Accession Numbers

The raw NGS reads were deposited in the database of Genbank (PRJNA639353). The GenBank accession numbers for the novel viral genomes described here are MT649483 and MT649484 (novel statovirus), MT649485 (novel cyclovirus), and MT649486 (novel gemycircularvirus).

### 2.10. Statistics

Pearson’s Chi-squared test or Fisher exact test or *t*-test was applied in the calculation of associations or differences between variables by pairwise comparisons. The *p* values were adjusted for multiple comparisons using the Benjamini and Hochberg method [25] with a false discovery rate (FDR) calculator [26]. A *p* ≤ 0.05 was considered significant. The Wilson method in EpiTools [27] was used to calculate 95% confidence intervals. Pearson’s correlation coefficients were calculated by STATA software 12.0 to measure the correlation of normally distributed variables. The normality was tested with normal Q-Q plots by STATA.

## 3. Results

### 3.1. Characteristics of the Cohort Members and Clinical Samples

We first applied mNGS to nasal-throat swab samples collected during disease episodes in 2013. These samples consisted of 94 samples from 94 disease episodes from 60 study participants residing in Dong Thap province. A convenience sample size of 91 samples from 58 individuals was selected from these 94 samples of 60 participants for metagenomics analysis. Of these, 15 were positive for at least one respiratory viral pathogen and were included as positive controls for mNGS analysis alongside the remaining 76 RT-PCR negative swabs. The demographic and baseline characteristics of the participants are presented in Table 1.

### 3.2. Overview of Sequences Detected by Metagenomics

A total of 31,783,202 raw reads were obtained, with a median read of 342,524 and range of 43,930–718,762 reads/sample. Most of the reads were ~145–150bp length. Reads belonged to viral, bacterial, and human as well as unclassifiable sequences. We focused on viral reads from eukaryotic viruses, which in total accounted for 2.3% (range: 0.5–12.7%) of the total reads obtained from individual samples. 

Evidence of the sequences related to 52 viral species from 31 families (including 19 viral species from 13 families that have previously been reported in human samples) was found in 27 of 91 (29.7%, 95% confidence interval (CI): 21.3–39.7%) samples but not in the negative control. After confirmatory PCR for a subset of viruses, the presence of 12 virus species from 9 families could be confirmed in 22 of 91 samples (24.2%, 95% CI: 16.5–33.9%) (Table 2 and Table 3).

Sequences related to those of viruses of invertebrates, plants, fungi, algae, and bacteria were also detected (Appendix A). 

### 3.3. Viral Detection in Positive Controls

Sequences related to 5 viral pathogens detected by diagnostic RT-PCRs were detected by mNGS in 13 of 15 (86.7%, 95% CI: 62.1–96.3%) samples, including 8 human rhinovirus (HRV), 1 enterovirus (EVs), 1 mix-detection of HRV and EVs, 1 influenza A virus, 1 coronavirus (CoV), and 1 respiratory syncytial virus (RSV) A. mNGS failed to detect RSV (cycle threshold (Ct): 36.3) and human metapneumovirus (MPV) (Ct: 40) in 2/15 RT-PCR-positive samples (Table 2) but detected in only 2 other samples, which were negative in RT-PCR (Table 3).

Using the mNGS sequences, the viruses detected in 13/15 patients by diagnostic RT-PCRs were successfully genotyped (Table 2). Based on mNGS sequences, cross-detection between EVs and HRV by RT-PCRs in two samples was detected and corrected to HRV-B and EV-D68, respectively (Table 2), and generated (almost) complete genomes (≥75% coverage) of HRV (*n* = 6) and EVs (*n* = 2) (Table 2). Phylogenetic analysis of the six HRV sequences revealed that they belong to species B (*n* = 4), species A (*n* = 1) and species C (*n* = 1) (Appendix A).

### 3.4. Viral Detection in RT-PCR-Negative Swabs and Results of Confirmatory PCR

Of the 76 RT-PCR-negative samples, sequences related to 12 viral species of 9 families that have previously been reported in both sterile and non-sterile human samples were found in 16 of 76 (21.1%) samples. They included both known human viruses (rotavirus, MPV, RSV, torque teno virus, human papillomavirus) and other viruses whose tropism is still unknown (novel cyclovirus, novel gemycircularvirus, novel statovirus, viruses of the *Circoviridae* family, gemycircularvirus, and statovirus) (Table 3). 

In the 15 control samples where human viral pathogens were previously detected by diagnostic RT-PCR, we also detected by mNGS the following viruses: human betaherpesvirus 7, vientovirus, gemycircularvirus, bat badicivirus-like virus [28], and bat posalivirus-like virus [28] in 4 of 15 (26.7%) samples (Table 2 and Table 3). 

Using specific PCR, we were able to confirm the presence of rotavirus (*n* = 1), novel cyclovirus (*n* = 1), novel gemycircularvirus (*n* = 3), novel statovirus *n* = 3), gemycircularvirus (*n* = 3), statovirus (*n* = 2), and vientovirus (*n* = 1) (Table 3) in 10 samples (8 of 76 (10.5%) RT-PCR-negative and 2 of 15 (13.3%) RT-PCR-positive samples).

### 3.5. Detection and Genomic Characterization of Novel Viruses

#### 3.5.1. A Novel Cyclovirus

Two cyclovirus-related contigs were generated from the NGS output of a single sample of acute respiratory disease with an unknown (diagnostic RT-PCR negative) etiology. A complete circular DNA genome of 1740 bp was then obtained by inverse PCR and Sanger sequencing (Appendix A). The complete genome shared the highest nucleotide identity (55%) to cyclovirus NG 14 (accession number: NC_038417), which is lower than the species demarcation threshold (80% identity of the genome-wide nucleotide sequence) [29]. Phylogenetic analysis of capsid and rep proteins, 216 and 279 amino acids long, respectively, confirmed its genetic distinction from other cycloviruses (Figure 2), suggesting that it is a novel cyclovirus species, tentatively named Cyclovirus VIZIONS-2013 (CyCV-VZ13). 

Besides being detected in 1 of 91 samples by mNGS analysis, subsequent PCR screening with primers based on mNGS contigs detected the CyCV-VZ13 genome in 5 of the 90 mNGS-negative samples (5.6%). Mix detections with other viruses were found in several samples (Appendix A).

#### 3.5.2. A Novel Gemycircularvirus

Multiple gemycircularvirus-related contigs were detected in the mNGS output of a single sample of acute respiratory disease with a RT-PCR-unknown etiology. Based on PCR with inverse primers (Appendix A), and Sanger sequencing, a complete circular DNA genome of 2171 bp was generated. The complete genome shared the highest nucleotide identity (48.3%) to a murine feces-associated gemycircularvirus 2 (GenBank: MF416388.1). The species demarcation of gemycircularvirus is a 78% genome-wide pairwise identity [30]. All of the proposed species (*n* = 43; 73 strains) within the genus *Gemycircularvirus* share 56–77% whole-genome similarity with each other [30]. These suggest that a member of a novel gemycircularvirus species was discovered, which we tentatively named gemycircularvirus VIZIONS-2013 (GemyCV-VZ13). The phylogenetic analyses of the capsid (298 amino acid) and replication proteins (333 amino acid) were in agreement with this suggestion (Figure 3). 

The sequences of GemyCV-VZ13 were found in the mNGS output and confirmed by PCR of two more samples. Besides being detected in 3 of 91 samples in the mNGS analysis, subsequent PCR screening yielded evidence of GemyCV-VZ13 in 12 of 88 (13.6%) mNGS-negative samples. Mix detections with other viruses were also found (Appendix A). 

#### 3.5.3. A Novel Statovirus

Seven statovirus-related contigs were detected in mNGS output from three nasal-throat swab samples with negative diagnostic RT-PCR. Subsequently, partial RNA-dependent RNA polymerase (RdRp) and coat protein sequences were generated with 249 and 260 amino acids in length, respectively, sharing 40.4% and 45% amino acid identity with available statoviruses sequences. Currently, no species/genus demarcation of statoviruses is available [31]; however, based on the low identity of the RdRp protein sequence with other statoviruses and the distinction in the phylogenetic tree (Figure 4), we proposed this as a novel statovirus species, tentatively named statovirus VIZIONS-2013 (StatoV-VZ13). 

Besides being detected in 3 of 91 nasal-throat swab samples by mNGS, additional PCR screening with primers based on mNGS contigs did not detect StatoV-VZ13 in any of the mNGS-negative samples. Thus, StatoV-VZ13 was detected in 3/91 (3.3%) samples collected at disease episodes.

## 4. Discussion

We describe here viral nucleic acids in nasal-throat swab samples from cases of acute respiratory diseases of unknown etiology from people at risk of zoonotic infections from Dong Thap province of Vietnam in 2013. We identified 12 species from 9 families of viruses that have previously been reported in various human samples. Sequences related to bacterial viruses, invertebrate viruses, fungal viruses, insect viruses, plant viruses, and algae viruses were also detected in the samples. Therefore, this viral survey expands our understanding of virus populations in acute respiratory diseases, particularly in people at risk of zoonotic infections, in Vietnam.

Metagenomic detection of most respiratory viral pathogens detected by RT-PCR indicated that the mNGS pipeline/protocol applied here is a sensitive pan-pathogen assay of respiratory viral pathogens in clinical samples, in agreement with previous studies [6,17,32]. Only sequences of RSV A and MPV were not detected from metagenomic output in the two diagnostic RT-PCR positive samples (RSV A- and MPV-PCR-positive samples) but were instead identified in only two other samples, suggesting that index-hopping probably happened, although contamination or pipette mistakes were not excluded. Additionally, detecting and correcting cross-reactivity between EVs and HRV in RT-PCR results, and genotyping other strains highlights one of the advantages of metagenomics in etiological and epidemiological studies compared to RT-PCR. Indeed, the analysis of HRV based on the obtained sequences suggested the predominance of HRV B in acute respiratory diseases in adults and imported HRV into Vietnam from several independent events. Notably, few studies reported the genetic diversity of HRV circulating in Vietnam. As such, our data has also shed light on the diversity of HRV in this locality.

The detection of several novel or recently identified viral genomes, including CyCV-VZ13, GemyCV-VZ13, StatoV-VZ13, and vientovirus, show that metagenomics is suitable as a sensitive pan-pathogen assay for sequence-independent detections of a variety of viruses, including novel ones. However, more frequent detections of the novel viruses by PCR than by metagenomics on the same samples suggests that PCR currently remains the most sensitive test for the diagnosis of those viruses whose genomes are already known. The main advantages of metagenomics are therefore the ability to detect and sequence all viral genomes simultaneously rather than perform an extensive set of different RT-PCRs. Currently, there are no robust criteria that can reliably define a true positive metagenomic result without the requirement of conducting confirmatory experiments [8,33]. As an exploratory study, we pragmatically took into account short viral reads presenting in the tested samples at any frequency for subsequent confirmatory PCR testing. This led to the discovery of several new viruses (CyCV-VZ13, StatoV-VZ13, and GemyCV-VZ13) in the present study, and the correct detection of influenza A virus in an RT-PCR-positive nasal-throat swab. Of note, a novel cyclovirus has previously been discovered and characterized based on a single initial read [34]. Collectively, the data thus suggest that even a single or a few viral reads generated by metagenomics can be a reliable marker for pathogen detection and discovery provided that the sequence similarity is high enough or used as an initial step towards generating a longer contig.

Cycloviruses belong to the *Circoviridae* family The closely related circoviruses are well known as pathogens in swine and birds and several other animals [35,36]. The natural hosts and pathogenic potentials of members of the *Cyclovirus* genus have not been definitely determined [36]. However, cyclovirus sequences have been detected in blood [37], cerebrospinal fluid [34,38], human respiratory tract [34,39], and persistent detection of identical sequences in the serum of immunodeficient patients [40]. Similarly, whether gemycircularviruses can infect humans is unknown. The germycircularvirus genome was identified in a wide range of host, in the feces of different animals, plants, insects, sewage, in the human respiratory tract [41], in blood from a patient with multiple sclerosis, and in the cerebrospinal fluid of encephalitis patients [41,42,43,44,45]. Statoviruses belong to a novel taxon of RNA viruses and have been detected in stool samples of diverse mammals, including human, macaque, mouse, and cow, but not in public sequencing datasets from bacteria, fungi, plants, unicellular eukaryotic organisms, or environmental samples and in the human respiratory tract [31]. The identification of novel viruses, including CyCV-VZ13, GemyCV-VZ13, and StatoV-VZ13, contributes to a better understanding of the respiratory virome in this part of the world. However, sequences related to viruses of the phylum Cressdnaviricota are ubiquitous contaminants of commonly used metagenomic reagents [46,47]. Thus, whether these genomes infect human cells, other non-human cells in the lungs, or reflect passive contamination of the respiratory tract will require further studies.

## 5. Conclusions

Our study demonstrates the presence of known and novel viruses in patients with acute respiratory diseases at risk of zoonotic infections. mNGS is a sensitive pan-pathogen assay for sequence-independent detection of respiratory viral pathogens in clinical samples. The detection of several novel viruses further contributes to our understanding of the human respiratory virome, and warrants further research to ascribe the clinical significant potential of these novel viruses. 

## Figures and Tables

**Figure 1 viruses-12-00960-f001:**
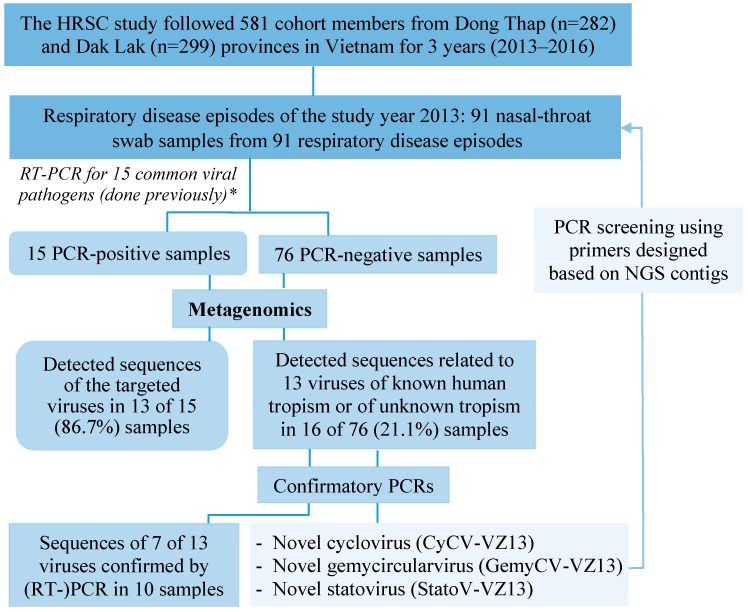
Overview of the methods and main results. HRSC: high risk sentinel cohort. * see [5], the 15 viruses include human rhinovirus (HRV), enterovirus (EVs), coronavirus (CoV) subtype OC43 and NL63, respiratory syncytial virus (RSV) A, RSV B, human metapneumovirus (MPV), influenza A virus, influenza B virus, adenovirus, parainfluenza virus (PIV)1–4, human bocavirus, and parechovirus.

**Figure 2 viruses-12-00960-f002:**
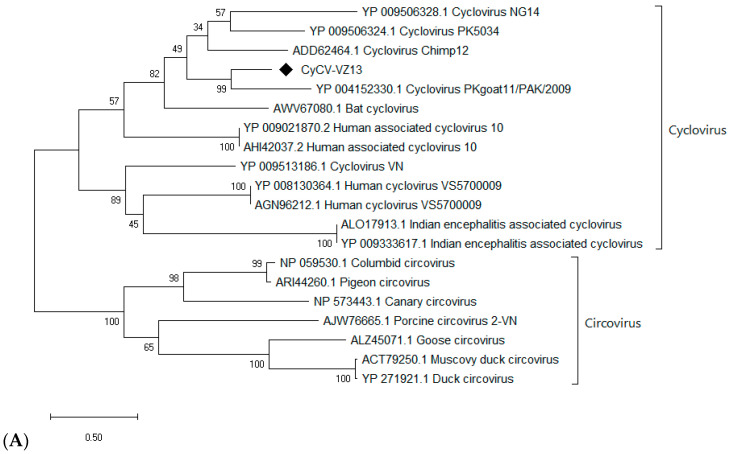
Phylogenetic tree of capsid (**A**) and replication (**B**) protein of Cyclovirus VIZIONS-2013 (CyCV-VZ13) compared to known viruses of the *Circoviridae* family.

**Figure 3 viruses-12-00960-f003:**
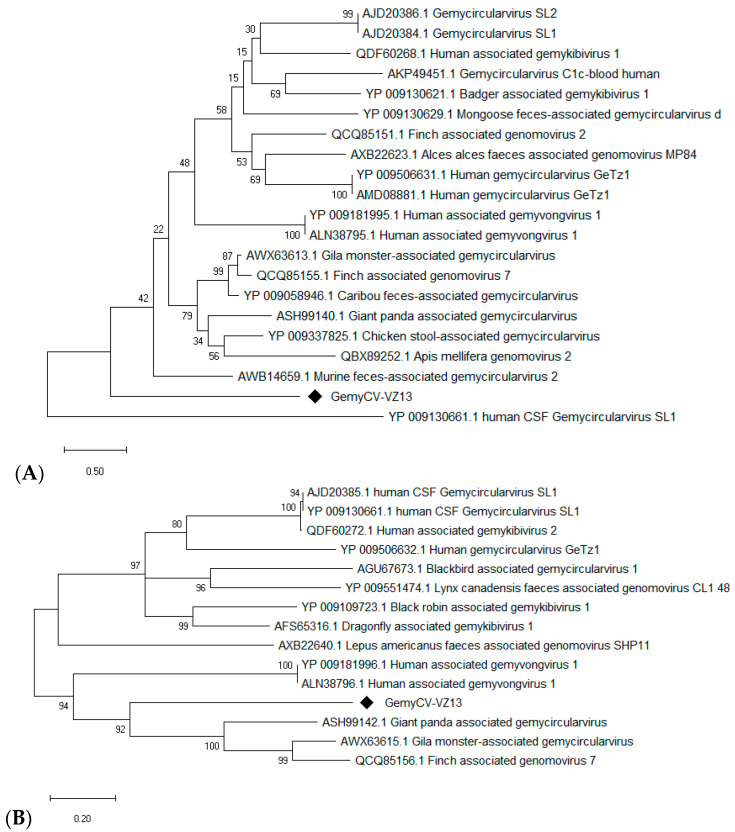
Phylogenetic tree of capsid (**A**) and replication (**B**) proteins of the gemycircularvirus VIZIONS-2013 (GemyCV-VZ13) compared to the viruses of the *Genomoviridae* family.

**Figure 4 viruses-12-00960-f004:**
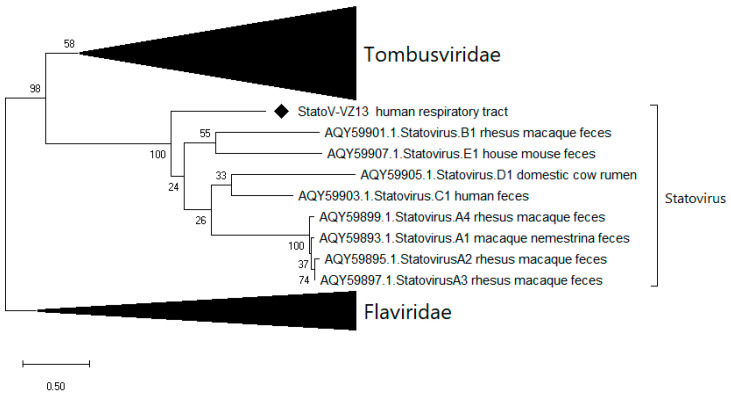
Phylogenetic tree of 249 amino acid partial RNA-dependent RNA polymerase (RdRp) protein sequences of statovirus VIZIONS-2013 (StatoV-VZ13) compared to the statoviruses on GenBank and viruses of the Tombusviridae and *Flaviridae* family.

**Table 1 viruses-12-00960-t001:** General characteristics of the 58 cohort members, and comparison between clinical symptoms recorded at respiratory-disease episodes of PCR-positive and -negative individuals.

	mNGS Analysis			
	Total	PCR Positive	PCR Negative	*p* Value ^#^^
**No. of cohort members**	***N* = 58**	***N* = 14**	***N* = 51**	
**Median age** (range) (in years)	35.5 (7–76)	31 (13–58)	38 (7–76)	0.465 ^##^
**Sex ratio** (male/female)	2.6 (42/16)	1.3 (8/6)	3.3 (39/12)	0.465
**Occupations**				
Animal health worker	12 (20.7)	2 (14.3)	11 (21.6)	1
Animal-raising farmer	26 (44.8)	6 (42.9)	22 (43.1)	1
Slaughterer	18 (31.0)	5 (35.7)	17 (33.3)	1
Rat-trader	2 (3.4)	1 (7.1)	1 (2.0)	0.829
**Having chronic diseases**	4 (6.9)	0	4 (7.8)	1
**Respiratory disease episodes**	***N* = 91**	***N* = 15**	***N* = 76**	
Frequency of clinical signs				
Fever	91 (100)	15 (100)	76 (100)	-
Cough	75 (82.4)	8 (53.3)	67 (88.2)	**0.015**
Sneezing	69 (75.8)	14 (93.3)	55 (72.4)	0.465
Sore throat	49 (53.8)	8 (53.3)	41 (53.9)	1
Dyspnea	9 (9.9)	1 (6.7)	8 (10.5)	1
Headache	57 (62.6)	12 (80)	45 (59.2)	0.465
Body aches	47 (51.6)	7 (46.7)	40 (52.6)	1
Watery diarrhea	11 (12.1)	0 (0)	11 (14.5)	0.5
Nausea	2 (2.2)	0 (0)	2 (2.6)	1

The value shows in format of number (percentage). ^#^ between PCR-positive vs. PCR-negative columns conducted by Pearson’s Chi-squared test or Fisher exact test. ^##^ by *t*-test. ^ the *p* values of multiple comparisons were corrected by the Benjamini and Hochberg method for false discovery rate (FDR) correction.

**Table 2 viruses-12-00960-t002:** Detection of respiratory viral pathogens of mNGS in 15 RT-PCR-positive samples where human viral pathogens were previously detected by diagnostic RT-PCR [5]. HRV: human rhinovirus, EVs: enterovirus, CoV: coronavirus subtype OC43 and NL63, RSV: respiratory syncytial virus, MPV: human metapneumovirus.

No.		Multiplex RT-PCR **		NGS Analysis				
	Sample ID	Virus Detected	Ct Value	Virus Genotype	Reads (%) ^#^	Total Length (bp)	Genome Coverage (%)	Other Virus Detected ^##^
1	72	EVs	32.4	Coxsackievirus A21	52,989 (12)	7440	100.0	
		HRV	37.1	HRV C56	2506 (0.6)	7099	98.1	
2	75	EVs	38.6	HRV B	4 (0.0)	598	8.3	
3	5	HRV	38.4	HRV B3	678 (0.7)	5512	75.0	Human betaherpesvirus 7
4	33	HRV	40	EVs-D68	3174 (0.7)	5629	76.2	
5	54	HRV	40	HRV B	6 (0.0)	723	10.0	
6	73	HRV	40	HRV B86	6644 (1.5)	7212	99.2	Vientovirus
7	83	HRV	38.7	HRV B79	6157 (1.8)	5639	78.2	Novel gemycircularvirus (GemyCV-VZ13)
8	86	HRV	38.2	HRV B79	19,606 (5.6)	7224	99.7	
9	91	HRV	40	HRV A57	2538 (1.1)	3450	47.8	
10	92	HRV	36.5	HRV B35	12,481 (3.1)	7298	100.0	Bat badicivirus, bat posalivirus
11	4	Influenza A virus	29.3	Influenza A/N2 virus	2 (0.0)	115	0.8	
12	6	CoV *	36	CoV OC43	8 (0.0)	733	2.4	
13	52	RSV-A	30.8	RSV-A genotype ON1	236 (0.1)	5398	35.4	
14	39	RSV-A	36.3	Not detected	0	0	0	
15	65	MPV	39.5	Not detected	0	0	0	

* OC43 or/and NL63. **** reported previously [5]. ^#^ Total reads of the targeted virus (percentage: the total reads of the virus per total raw reads of the sample). ^##^ detail of the viruses in Table 3.

**Table 3 viruses-12-00960-t003:** Metagenomic detection of viruses that have previously been detected in human samples in nasal-throat swab samples negative for human viral pathogens by diagnostic RT-PCR [5].

No.	Sample ID	Detected Viruses Previously Reported in Human Samples	Confirmed by PCR	No. of Reads	Total Contig Length (bp)	Amino Acid Identity to GenBank Strain (%)	Genome Coverage (%)
1	89	Rotavirus	Yes	17	360	98	1.9
2	73	Vientovirus *^#^	Yes	2	146	53	4.8
3	23	Novel cyclovirus (CyCV-VZ13)	Yes	5	448	61.8	25.9
4	32	Novel gemycircularvirus virus (GemyCV-VZ13)	Yes	1852	1995	39	91
5	83	GemyCV-VZ13	Yes	120	2000	45	92
6	89	GemyCV-VZ13	Yes	1	148	46.9	6.8
7	24	Novel statovirus (StatoV-VZ13)	Yes	91	1018	42.5	24.6
8	32	StatoV-VZ13	Yes	5	231	35	5.6
9	82	StatoV-VZ13	Yes	27	2000	49	48.4
10	87	Gemycircularvirus	Yes	39	858	83	39
11	71	Gemycircularvirus	Yes	117	1400	97	63.7
12	88	Gemycircularvirus	Yes	2	300	73	13.6
13	11	Statovirus	Yes	4	351	91	8.5
14	71	Statovirus	Yes	7	812	90	19.6
15	5	Human betaherpesvirus 7 *	Not done	2	295	100	0.2
16	15	Human papillomavirus	Not done	73	1280	99.3	17.5
17	17	Human papillomavirus	Not done	6	437	97.9	6
18	2	Torque teno virus	Not done	4	554	88.4	14.6
19	68	Torque teno virus	Not done	2	217	70.6	5.7
20	24	MPV	No	6	417	100	3.1
21	47	RSV A	No	6	468	100	3.1
22	92	Bat badicivirus-like virus *	No	2	204	49	2.3
23	92	Bat posalivirus-like virus *	No	3	182	56	2
24	83	Viruses of *Circoviridae* family	No	10	167	64	7

* Co-detected with other viral pathogens in 15 positive-control samples as reported in Table 2. ^#^ will be described in a separate paper.

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
