# Peer review of "The Virome of Acute Respiratory Diseases in Individuals at Risk of Zoonotic Infections"

_viruses, 2020, doi:10.3390/v12090960_

Round 1

Reviewer 1 Report

The proposed manuscript describes an interesting work about the potential of NGS in the diagnostic field, in particular the metagenomics sequencing.

Despite the potential of the work and presented results, bioinformatics methods to produce such results are not enough precise to exclude false positives and assess real presence of the certain viruses, thus making less meaningful the comparison with classical diagnostic test like RT-PCR. I strongly suggest to redo the bioinformatic analysis by adding more stringent measures (like a minimum number of reads) and running de novo assembly with different settings, softwares and pooled reads.

In particular, some results detailed in paragraph 3.3 e table 2 are not acceptable; detecting the presence of a virus based on only 2 reads (e.g. influenza A virus) is not believable, since Illumina sequencing suffer from the cross contamination between samples, due to the impossibility to achieve 100% of purity for indexes used to multiplex samples.
Moreover, the limit of 40 for Ct in a RT-PCR in only theoretical, and values over 35 are most of the times false positives. These results should be revised in order to present only the reliable ones.

I also have some suggestions on minor changes to do:

  • Line 110: Could you please detail the composition of the non-redundant non-virus protein database used? And also please detail the threshold values for filtering out false-positive reads.
  • Line 112: It is not clear why did you use blastx after DIAMOND: the latter is a faster implementation of the blastx algorithm, so blastx is not expected ot produce alignments significant different from those produced by DIAMOND...could you expand this method in order to explain better?
  • Line 132: Against which database were performed the blastx alignment?
  • Line 177-187: This is the same identical senteces of lines 165-175. Please avoid repetitions.
  • Line 202: 12 out of 15 samples have Ct >35, 9 out of 15 samples have Ct >38: more that half viral detection in mNGS has totally unreliable Ct values. This is not a successfull genotyping!
  • Figure 2: despite the high R values obtained for the correlations, the figure shows that the correlation test used is heavy influenced by the presence of a single value around 32 Ct for the first part of the correlation line and all the remaining values at Ct>36 for the second part of the correlation line. So the result of correlation test is very biased.
  • Table 3: same problems in paragraph 3.3; too many few reads to assess the presence of a virus in a sample. Also is not believable that 2 reads of 300bp length (or nearly) assemble togheter in a contig of 146 bp. Finally, an aminoacidid identity of only 40% is not enough to classify a contig as belonging to a particular virus family/species

Author Response

August 19th, 2020

Dear Reviewer 1,

We would also like to thank you for critical reviewing of our paper and constructive comments. We read with interest your comments, which have certainly helped to improve our manuscript. Please find enclosed our point-by-point response, addressing your comments.  

-----------------------------------------------------------------------------------

The proposed manuscript describes an interesting work about the potential of NGS in the diagnostic field, in particular the metagenomics sequencing.

Point 1: Despite the potential of the work and presented results, bioinformatics methods to produce such results are not enough precise to exclude false positives and assess real presence of the certain viruses, thus making less meaningful the comparison with classical diagnostic test like RT-PCR. I strongly suggest to redo the bioinformatic analysis by adding more stringent measures (like a minimum number of reads) and running de novo assembly with different settings, softwares and pooled reads.

In particular, some results detailed in paragraph 3.3 e table 2 are not acceptable; detecting the presence of a virus based on only 2 reads (e.g. influenza A virus) is not believable, since Illumina sequencing suffer from the cross contamination between samples, due to the impossibility to achieve 100% of purity for indexes used to multiplex samples.

Moreover, the limit of 40 for Ct in a RT-PCR in only theoretical, and values over 35 are most of the times false positives. These results should be revised in order to present only the reliable ones.

Response 1: The bioinformatics approaches (including the de novo assembly program) used are now extensively described (reference #18). The performance of our de novo assembly method is comparable and often superior to other de novo assembly program.

Cross-contaminations does commonly occur in metagenomics. However, only one sample was positive for influenza A virus using RT-PCR prior to analysis by metagenomics. Metagenomics did find 2 reads of influenza A virus in that same (RT-PCR positive) sample only. Additionally, influenza A virus sequences were not found in the remaining samples (including the negative control). We therefore conclude that the presence of influenza A was supported by two assays and was not the result of bleedover as its presence was confirmed by RT-PCR (Ct 29).

Collectively, in response to your other comments, we have revised the data analysis section (section 2.4) to further elaborate on our approaches as below.

“2.4. Analysis of mNGS sequence data

An in-house analysis pipeline was used to analyze sequence data that is posted in GitHub: https://github.com/xutaodeng/virushunter/. Briefly, the adaptors, low quality reads, duplicate reads were firstly removed. The reads related to human and bacterial genomes were subtracted by mapping reads using bowtie2 (version 2.2.4) to concatenated human reference genome sequence and mRNA sequences (hg38), and bacterial nucleotide sequences extracted from NCBI nt fasta file (ftp://ftp.ncbi.nlm.nih.gov/blast/db/FASTA/, February 2019) based on NCBI taxonomy (ftp://ftp.ncbi.nih.gov/pub/taxonomy, February 2019) [18]. The remaining reads were de novo assembled using ENSEMBLE software that uses a partitioned subassembly approach to integrate the use of various de Bruijn graph (DBG) and overlap-layout-consensus assemblers (OLC) [19]. To allow for sensitive screening of viral sequences, the resulting contigs (plus single reads) were aligned against the viral proteome of the NCBI’s RefSeq and the viral proteome of the non-redundant database by Basic Local Alignment Search Tool (BLASTx). Matches with E score <0.01 were retained. To filter out tentative viral hits that showed better alignments to non-viral sequences these tentative matches to viral proteins were then aligned to the GenBank’s entire non-redundant proteome database using DIAMOND algorithm version 0.9.6 [20]. Sequences were then classified as viral or removed as non-viral according to NCBI taxonomy of the best hits (lowest E score) in the non-redundant proteome database. Viral reads described here have E scores to viral proteins <10-10.”

A Ct value cut off of ≤40 has previously been applied (PMID: 21571585, https://doi.org/10.1371/journal.pone.0143164). The former is the assay that we used to screen our respiratory samples as described elsewhere (PMID: 31769525). Additionally, as the reviewer appreciates, Ct value might vary between PCR platforms. Indeed, despite high Ct value (low viral load), high genome coverage was successfully generated by mNGS (Table 2).

Last but not least, as the reviewer appreciates, currently, there are no robust criteria that can reliably define a true positive metagenomic result without the requirement of conducting confirmatory experiments. As an exploratory study, we thus pragmatically took into account viral reads presenting in the tested samples at any frequency for subsequent confirmatory PCR testing. Therefore, we have added some text to the discussion section; line 358-367 reads “Currently, there are no robust criteria that can reliably define a true positive metagenomic result without the requirement of conducting confirmatory experiments [8,33]. As an exploratory study, we pragmatically took into account short viral reads presenting in the tested samples at any frequency for subsequent confirmatory PCR testing. This has led to the discovery of several new viruses (CyCV-VZ13, StatoV-VZ13 and GemyCV-VZ13) in the present study, and the correct detection of influenza A virus in an RT-PCR positive nasal-throat swab. Of note, a novel cyclovirus has previously been discovered and characterized based on a single initial read [34]. Collectively, the data thus suggest that even a single or a few viral reads generated by metagenomics can be a reliable marker for pathogen detection and discovery provided that the sequence similarity is high enough or used as an initial step towards generating a longer contig”

Point 2: I also have some suggestions on minor changes to do:

Line 110: Could you please detail the composition of the non-redundant non-virus protein database used? And also please detail the threshold values for filtering out false-positive reads.

Response 2: The database used was “GenBank’s entire non-redundant proteome” (line 125-126) with non-viral sequences identified based on taxonomy annotation. The best E value of <10-10 to viral sequences was applied as the threshold value. Furthermore, in response to your other comments, we have now added more details to section 2.4 to further describe our methodology (please refer to the newly added text in section 2.4. mentioned as above).  

Point 3: Line 112: It is not clear why did you use blastx after DIAMOND: the latter is a faster implementation of the blastx algorithm, so blastx is not expected ot produce alignments significant different from those produced by DIAMOND...could you expand this method in order to explain better?

Response 3: We first used BLASTx (rather than DIAMOND) against viral proteins from RefSeq and non-redundant database as a slightly more sensitive if slower tool to identify potential viral sequences. We then took these tentative hits and ran them against the much larger entire non-redundant proteome database using the faster DIAMOND program to remove those reads/contigs with better match to non-viral than to viral sequences. The revised text now reads (line 120-128). Please also refer to section 2.4. and our response to your other comments related to our bioinformatics approaches.  

Point 4: Line 132: Against which database were performed the blastx alignment?

Response 4: We revised the manuscript based on your comments, line 120-123 now reads: “To allow for sensitive screening of viral sequences, the resulting contigs (plus single reads) were aligned against the viral proteome of the NCBI’s RefSeq and the viral proteome of the non-redundant database by Basic Local Alignment Search Tool (BLASTx).”

Point 5: Line 177-187: This is the same identical senteces of lines 165-175. Please avoid repetitions.

Response 5: We deleted the repetition (lines 189-199) based on your comments.

Point 6: Line 202: 12 out of 15 samples have Ct >35, 9 out of 15 samples have Ct >38: more that half viral detection in mNGS has totally unreliable Ct values. This is not a successfull genotyping!

Response 6: Please refer to our response to your comment #1 above. Notably, high genome coverage >47% was obtained in 9/15 were a viral genotype/serotype was successfully defined (Table 2).  

Point 7: Figure 2: despite the high R values obtained for the correlations, the figure shows that the correlation test used is heavy influenced by the presence of a single value around 32 Ct for the first part of the correlation line and all the remaining values at Ct>36 for the second part of the correlation line. So the result of correlation test is very biased.

Response 7: We agree with the reviewer that our analysis was potentially biased due to the small sample size with a narrow interval of viral loads. With that in mind, we have now removed the corresponding correlation result and the related discussion.

Point 8: Table 3: same problems in paragraph 3.3; too many few reads to assess the presence of a virus in a sample. Also is not believable that 2 reads of 300bp length (or nearly) assemble togheter in a contig of 146 bp. Finally, an aminoacidid identity of only 40% is not enough to classify a contig as belonging to a particular virus family/species.

Response 8: Most of the reads are ~145-150bp length, so 2 reads assembled together to make a contig of 146 bp. We added this point to the manuscript, line 203 reads “Most of the reads were ~145-150bp length”.

As for viral species assignment based on the sequence similarity at amino acid level, we considered these as sequences that potentially belonged to a novel viruses, and thus conducted further experiments to further characterize them. This has led to the discovery of the novel viruses presented in the manuscript. Please refer to Figure 1 and the method sections (section 2.5 and 2.6) where we presented our approaches to further investigate the obtained sequences that potentially belonged to a novel virus.

We have added a paragraph to the discussion section as per the reviewer’s suggestion; line 358-367 reads “Currently, there are no robust criteria that can reliably define a true positive metagenomic result without the requirement of conducting confirmatory experiments [8,33]. As an exploratory study, we pragmatically took into account short viral reads presenting in the tested samples at any frequency for subsequent confirmatory PCR testing. This has led to the discovery of several new viruses (CyCV-VZ13, StatoV-VZ13 and GemyCV-VZ13) in the present study, and the correct detection of influenza A virus in an RT-PCR positive nasal-throat swab. Of note, a novel cyclovirus has previously been discovered and characterized based on a single initial read [34]. Collectively, the data thus suggest that even a single or a few viral reads generated by metagenomics can be a reliable marker for pathogen detection and discovery provided that the sequence similarity is high enough or used as an initial step towards generating a longer contig.”

We hope you agree with our revisions and will now find our manuscript suitable for publication. 

Reviewer 2 Report

In this manuscript, Nguyen et al characterize the nasopharyngeal virome of patients working with animals and presenting acute respiratory diseases. The objective of this study is very ambitious as it could lead to identification of novel emerging zoonotic viruses. However this study has a significant flaw: it does not include any negative control to identify which viral sequences were present in laboratory reagents and potential cross-contamination between samples.

Metagenomics is a highly sensitive and powerful approach but it is also highly susceptible to inaccurate conclusions resulting from unintended sequencing of contaminants, especially those present in molecular biology reagents. For instance please see PMID: 25387460 on the impact of contaminants on metagenomics results; and PMID: 31059795 which lists contaminating viral sequences present in viral metagenomics reagents. In particular, this paper reports that Circoviridae, and ssDNA viruses in general, are ubiquitous contaminants of commonly used laboratories reagent. This questions the relevance of the novel ssDNA viruses described in the submitted manuscript.

In addition, cross-contamination of viral sequences between samples can happen during initial extraction, dsDNA synthesis, library preparation or sequencing. The fact that the same rare viruses (novel GemyCV-VZ13, or Novel StatoV-VZ13) are found in 3 consecutive samples, with first sample having highest number of reads could be indicative of cross-contamination.

The authors confirm their finding by specific PCR but this does not rule-out cross-contamination arising during initial sample extraction.

Similarly, confirmation by PCR does not rule out contamination of reagents present in extraction reagents or laboratory environment.

Therefore it is crucial in metagenomics to include several negative controls at each step of the process, including the initial extraction of the samples.

A second major points of criticism to be made is the identification of novel viruses using very short contigs and poor amino-acid identity. For instance a novel vientovirus is identified with 2 reads mapping a 146 bp contig with 53% AA identity to Genbank strain. How reliable are such prediction?

Author Response

August 19th, 2020

Dear Reviewer 2,

We would also like to thank you for critical reviewing of our paper and constructive comments. We read with interest your comments, which have certainly helped to improve our manuscript. Please find enclosed our point-by-point response, addressing your comments.  

-----------------------------------------------------------------------------------

Point 1: In this manuscript, Nguyen et al characterize the nasopharyngeal virome of patients working with animals and presenting acute respiratory diseases. The objective of this study is very ambitious as it could lead to identification of novel emerging zoonotic viruses. However this study has a significant flaw: it does not include any negative control to identify which viral sequences were present in laboratory reagents and potential cross-contamination between samples.

Metagenomics is a highly sensitive and powerful approach but it is also highly susceptible to inaccurate conclusions resulting from unintended sequencing of contaminants, especially those present in molecular biology reagents. For instance please see PMID: 25387460 on the impact of contaminants on metagenomics results; and PMID: 31059795 which lists contaminating viral sequences present in viral metagenomics reagents. In particular, this paper reports that Circoviridae, and ssDNA viruses in general, are ubiquitous contaminants of commonly used laboratories reagent. This questions the relevance of the novel ssDNA viruses described in the submitted manuscript.

The authors confirm their finding by specific PCR but this does not rule-out cross-contamination arising during initial sample extraction.

Similarly, confirmation by PCR does not rule out contamination of reagents present in extraction reagents or laboratory environment.

Therefore it is crucial in metagenomics to include several negative controls at each step of the process, including the initial extraction of the samples.

Response 1: Please accept our apologies for the confusion. We indeed included one negative-control sample (a viral transport medium samples) in our analysis. Accordingly, we have now revised the text to reflect this and the associated findings, line 94-95 now reads “Initially, 200µ of nasal-throat swabs collected at disease episodes and a negative control containing viral transport medium were first treated with 20U of turbo DNase”. And line 207-209 reads “Evidence of sequences related to 52 viral species from 31 families (including 19 viral species from 13 families that have previously been reported in human samples) was found in 27 of 91 (29.7%, 95% CI: 21.3–39.7%) samples, but not in in the negative control.”

We read with interest the two papers that the reviewer suggested. We thus added some text to further emphasize that care should be taken when interpreting metagenomic results (including those of the present study); line 382-386 reads “However, sequences related to viruses of the phylum Cressdnaviricota are ubiquitous contaminants of commonly used metagenomic reagents [47,48]. Thus, whether these genomes infect human cells, other non-human cells in the lungs or reflect passive contamination of the respiratory track will require further studies”

We found that paper 1 (PMID: 25387460) was concerned with bacterial sequences, which were not the focus of the present study. We thus did not cite this paper in our revised manuscript.

Point 2: In addition, cross-contamination of viral sequences between samples can happen during initial extraction, dsDNA synthesis, library preparation or sequencing. The fact that the same rare viruses (novel GemyCV-VZ13, or Novel StatoV-VZ13) are found in 3 consecutive samples, with first sample having highest number of reads could be indicative of cross-contamination.

Response 2:  As for the possibility of cross-contamination occurring during the laboratory procedures, we tried our best to minimize the possibility of contamination by performing specific PCR on newly extracted materials from the original samples (Section 2.5, line 139-140). Please also refer to our response above regarding the newly added text to elaborate on the uncertainty regarding the origin of novel viruses. 

Point 3: A second major points of criticism to be made is the identification of novel viruses using very short contigs and poor amino-acid identity. For instance a novel vientovirus is identified with 2 reads mapping a 146 bp contig with 53% AA identity to Genbank strain. How reliable are such prediction?

Response 3: Sequences related vientovirus will be reported in another paper (please refer to footnote of Table 3). As for viral species assignment based on the sequence similarity at amino acid level, we considered these as sequences that potentially belonged to a novel viruses, and thus conducted further experiments to further characterize them. This has led to the discovery of the novel viruses presented in the manuscript

We have added a paragraph to the discussion section as per the reviewer’s suggestion; line 358-367 reads “Currently, there are no robust criteria that can reliably define a true positive metagenomic result without the requirement of conducting confirmatory experiments [8,33]. As an exploratory study, we pragmatically took into account short viral reads presenting in the tested samples at any frequency for subsequent confirmatory PCR testing. This has led to the discovery of several new viruses (CyCV-VZ13, StatoV-VZ13 and GemyCV-VZ13) in the present study, and the correct detection of influenza A virus in an RT-PCR positive nasal-throat swab. Of note, a novel cyclovirus has previously been discovered and characterized based on a single initial read [34]. Collectively, the data thus suggest that even a single or a few viral reads generated by metagenomics can be a reliable marker for pathogen detection and discovery provided that the sequence similarity is high enough or used as an initial step towards generating a longer contig”

Additionally, we have revised the data analysis section (section 2.4) to further elaborate on our approaches as below.

“2.4. Analysis of mNGS sequence data

An in-house analysis pipeline was used to analyze sequence data that is posted in GitHub: https://github.com/xutaodeng/virushunter/. Briefly, the adaptors, low quality reads, duplicate reads were firstly removed. The reads related to human and bacterial genomes were subtracted by mapping reads using bowtie2 (version 2.2.4) to concatenated human reference genome sequence and mRNA sequences (hg38), and bacterial nucleotide sequences extracted from NCBI nt fasta file (ftp://ftp.ncbi.nlm.nih.gov/blast/db/FASTA/, February 2019) based on NCBI taxonomy (ftp://ftp.ncbi.nih.gov/pub/taxonomy, February 2019) [18]. The remaining reads were de novo assembled using ENSEMBLE software that uses a partitioned subassembly approach to integrate the use of various de Bruijn graph (DBG) and overlap-layout-consensus assemblers (OLC) [19]. To allow for sensitive screening of viral sequences, the resulting contigs (plus single reads) were aligned against the viral proteome of the NCBI’s RefSeq and the viral proteome of the non-redundant database by Basic Local Alignment Search Tool (BLASTx). Matches with E score <0.01 were retained. To filter out tentative viral hits that showed better alignments to non-viral sequences these tentative matches to viral proteins were then aligned to the GenBank’s entire non-redundant proteome database using DIAMOND algorithm version 0.9.6 [20]. Sequences were then classified as viral or removed as non-viral according to NCBI taxonomy of the best hits (lowest E score) in the non-redundant proteome database. Viral reads described here have E scores to viral proteins <10-10.”

We hope you agree with our revisions and will now find our manuscript suitable for publication. 

Reviewer 3 Report

Review report

Nguyen et al., 2020, The virome of acute respiratory disease in individuals with high risk of zoonotic infections

General comments

Metagenomic analyses of samples from respiratory disease patients working with animals. Application of a pipeline protocol for pan-viral detection in clinical samples. Very interesting cohort, straightforward methods.

There are some mistakes in the English writing. Please review present and past times in the verbs.

Specific comments

Title

The individuals tested are at risk of zoonotic infections but it was not really proven by the study that these people are at high risk. Better omit the word “high”. This needs to be extended to the whole the manuscript.

Abstract

The detected viruses are definitely part of the respiratory virome in the nasopharyngeal compartment. But how would you define “baseline virome”. This needs to be explained more.

M&M

Interesting metagenomics pipeline and bioinformatics approach. Clearly described protocol. Interesting cohort.

Results (incl. figures and tables)

Several new viruses were detected. But there is hardly any information in the manuscript about the role of these viruses. Are these viruses a steady part of the human virome in the respiratory tract in the studied region? Can they cause disease? Etc. What have the researchers done to address these questions?

Discussion

The title of the paper suggests the seach for zoonotic backgrounds of respiratory pathogens. However, a number of the viruses detected and listed are not really zoonotic. This needs to be explained in the discussion. What viruses are transimitted form animals to humans and what viruses have zoonotic background. In additinon, what does the work contribute to early detection of zoonotic transmission?

Overall conclusion

Interesting application of a viral metagenomics analysis pipeline. Nice combination of PCR and NGS use. Good example of an intereting cohort tested by novel molecular methods and bioinformatics analyses.

The paper needs some modifications but overall looks suitable to be accepted for publication in this journal.

Author Response

August 19th, 2020

Dear Reviewer 3,

We would also like to thank you for critical reviewing of our paper and constructive comments. We read with interest your comments, which have certainly helped to improve our manuscript. Please find enclosed our point-by-point response, addressing your comments.  

-----------------------------------------------------------------------------------

General comments

Metagenomic analyses of samples from respiratory disease patients working with animals. Application of a pipeline protocol for pan-viral detection in clinical samples. Very interesting cohort, straightforward methods.

Point 1: There are some mistakes in the English writing. Please review present and past times in the verbs.

Response 1: The manuscript has now been further reviewed and revised by a native English speaker who is a co-author of this manuscript.

Point 2: Specific comments

Title

The individuals tested are at risk of zoonotic infections but it was not really proven by the study that these people are at high risk. Better omit the word “high”. This needs to be extended to the whole the manuscript.

Response 2: We have removed the term “high” from the title and throughout the manuscript. Accordingly, the revised title now reads “The virome of acute respiratory diseases in individuals at risk of zoonotic infections”

Abstract

Point 3: The detected viruses are definitely part of the respiratory virome in the nasopharyngeal compartment. But how would you define “baseline virome”. This needs to be explained more.

Response 3: We agree with the reviewer that it was a confused phrase. We thus has removed the term baseline in the abstract and throughout the manuscript.

M&M

Interesting metagenomics pipeline and bioinformatics approach. Clearly described protocol. Interesting cohort.

Results (incl. figures and tables)

Point 4: Several new viruses were detected. But there is hardly any information in the manuscript about the role of these viruses. Are these viruses a steady part of the human virome in the respiratory tract in the studied region? Can they cause disease? Etc. What have the researchers done to address these questions?

Response 4: Please refer to the last paragraph of the discussion section where we discussed these aspects of the novel viruses.

Point 5: Discussion

The title of the paper suggests the seach for zoonotic backgrounds of respiratory pathogens. However, a number of the viruses detected and listed are not really zoonotic. This needs to be explained in the discussion. What viruses are transimitted form animals to humans and what viruses have zoonotic background. In additinon, what does the work contribute to early detection of zoonotic transmission?

Response 5: As the reviewer appreciates, discovery of a new virus is just the beginning of a new study. As such we proposed that additional work is essential to ascribe the clinical significant potentials of these novel viruses (line 390-392). We have also added some text to the conclusion section to elaborate on the requirement of further studies; line 390-392 reads “The detections of several novel viruses further contribute to our understanding of the human respiratory virome, and warrants further research to ascribe the clinical significant potential of these novel viruses.”

Overall conclusion

Interesting application of a viral metagenomics analysis pipeline. Nice combination of PCR and NGS use. Good example of an intereting cohort tested by novel molecular methods and bioinformatics analyses.

The paper needs some modifications but overall looks suitable to be accepted for publication in this journal.

Response 6: We hope you agree with our revisions and will now find our manuscript suitable for publication. 

Reviewer 4 Report

This manuscript reiterates the importance of metagenomic sequencing in clinical samples like many previous studies e.g. http://doi.org/10.1128/JCM.01641-14 and https://doi.org/10.1016/j.jcv.2015.06.082  

Potential typo in line 38 – “novel statovirus in none of metagenomics-negative samples”

Typo in line 111 – algorism should read algorithm

Text from section 3.2 is repeated at the end of section 3.1. Lines 165-175 are exactly same as lines 177-187.

Section 2.4 should include more detailed steps with relevant details where necessary as described below. It should be updated with details of different versions of tools and databases in for reproducibility purposes. In some cases, authors should also clarify the specific tools utilised e.g.  no information on de novo assembly tool provided.

“Briefly, the adaptors,low quality reads, duplicate reads and reads related to human

107 and bacterial genomes were firstly removed” - This step does not describe how were the reads removed e.g. by mapping entire set of sequences to reference genome using bwa or bowtie and extracting the unmapped reads or by BLASTing each read against the bacterial and human genomes?

Table 3 legend should be written more concisely. It would be great if the authors could add a section on identification of novel viruses from a few reads in discussion – strength of mNGS in relation to results described in table 3 in the manuscript. Additionally, table 3 shows that very short contigs (200bp) were subjected to downstream analysis and were later confirmed by PCR testing– it is unclear how post-assembly bioinformatic analysis was performed, did authors include all short contigs in the analysis? How authors have deemed them to be a real hit is unclear and should be included in the relevant section(s) in the manuscript.

Author Response

August 19th, 2020

Dear Reviewer 4,

We would like to thank you for critical reviewing of our paper and constructive comments. We read with interest your comments, which have certainly helped to improve our manuscript. Please find enclosed our point-by-point response, addressing your comments.  

-----------------------------------------------------------------------------------

This manuscript reiterates the importance of metagenomic sequencing in clinical samples like many previous studies e.g. http://doi.org/10.1128/JCM.01641-14 and https://doi.org/10.1016/j.jcv.2015.06.082 

Point 1: Potential typo in line 38 – “novel statovirus in none of metagenomics-negative samples”

Response 1: We revised the manuscript based on your comments that it is removed. Line 39-41 now reads: “Using PCR screening, the novel cyclovirus was additionally detected in 5 and the novel gemycircularvirus in 12 of the remaining samples included for mNGS analysis.”

Point 2: Typo in line 111 – algorism should read algorithm

Response 2: We revised the manuscript with “algorithm” taking over “algorism” (line 126, 156-158)

Point 3: Text from section 3.2 is repeated at the end of section 3.1. Lines 165-175 are exactly same as lines 177-187.

Response 3: We deleted the repetition in the manuscript based on your comments (line 189-199) 

Point 4: Section 2.4 should include more detailed steps with relevant details where necessary as described below. It should be updated with details of different versions of tools and databases in for reproducibility purposes. In some cases, authors should also clarify the specific tools utilised e.g.  no information on de novo assembly tool provided.

Response 4: Section 2.4 has now been substantially revised, which covered the areas that the reviewer commented on, to further elaborate on our approaches as below.

“2.4. Analysis of mNGS sequence data

An in-house analysis pipeline was used to analyze sequence data that is posted in GitHub: https://github.com/xutaodeng/virushunter/. Briefly, the adaptors, low quality reads, duplicate reads were firstly removed. The reads related to human and bacterial genomes were subtracted by mapping reads using bowtie2 (version 2.2.4) to concatenated human reference genome sequence and mRNA sequences (hg38), and bacterial nucleotide sequences extracted from NCBI nt fasta file (ftp://ftp.ncbi.nlm.nih.gov/blast/db/FASTA/, February 2019) based on NCBI taxonomy (ftp://ftp.ncbi.nih.gov/pub/taxonomy, February 2019) [18]. The remaining reads were de novo assembled using ENSEMBLE software that uses a partitioned subassembly approach to integrate the use of various de Bruijn graph (DBG) and overlap-layout-consensus assemblers (OLC) [19]. To allow for sensitive screening of viral sequences, the resulting contigs (plus single reads) were aligned against the viral proteome of the NCBI’s RefSeq and the viral proteome of the non-redundant database by Basic Local Alignment Search Tool (BLASTx). Matches with E score <0.01 were retained. To filter out tentative viral hits that showed better alignments to non-viral sequences these tentative matches to viral proteins were then aligned to the GenBank’s entire non-redundant proteome database using DIAMOND algorithm version 0.9.6 [20]. Sequences were then classified as viral or removed as non-viral according to NCBI taxonomy of the best hits (lowest E score) in the non-redundant proteome database. Viral reads described here have E scores to viral proteins <10-10.”

Point 5: “Briefly, the adaptors, low quality reads, duplicate reads and reads related to human

107 and bacterial genomes were firstly removed” - This step does not describe how were the reads removed e.g. by mapping entire set of sequences to reference genome using bwa or bowtie and extracting the unmapped reads or by BLASTing each read against the bacterial and human genomes?

Response 5: Section 2.4 has now been substantially revised to cover the areas that the reviewer commented on. Please refer to our response to your comment #4 above.

Point 6: Table 3 legend should be written more concisely. It would be great if the authors could add a section on identification of novel viruses from a few reads in discussion – strength of mNGS in relation to results described in table 3 in the manuscript. Additionally, table 3 shows that very short contigs (200bp) were subjected to downstream analysis and were later confirmed by PCR testing– it is unclear how post-assembly bioinformatic analysis was performed, did authors include all short contigs in the analysis? How authors have deemed them to be a real hit is unclear and should be included in the relevant section(s) in the manuscript.

Response 6: We have added a paragraph to the discussion section as per the reviewer’s suggestion; line 358-367 reads “Currently, there are no robust criteria that can reliably define a true positive metagenomic result without the requirement of conducting confirmatory experiments [8,33]. As an exploratory study, we pragmatically took into account short viral reads presenting in the tested samples at any frequency for subsequent confirmatory PCR testing. This has led to the discovery of several new viruses (CyCV-VZ13, StatoV-VZ13 and GemyCV-VZ13) in the present study, and the correct detection of influenza A virus in an RT-PCR positive nasal-throat swab. Of note, a novel cyclovirus has previously been discovered and characterized based on a single initial read [34]. Collectively, the data thus suggest that even a single or a few viral reads generated by metagenomics can be a reliable marker for pathogen detection and discovery provided that the sequence similarity is high enough or used as an initial step towards generating a longer contig.”

Please also refer to our response to the comment #4 regarding our bioinformatics approach to the detection of viral sequences in the metagenomic dataset. 

Finally, the legend of Table 3 has now been revised.

Round 2

Reviewer 1 Report

Overall, authors answered well to all the concerns I raised in the previous round of review. The word was well expanded and better described. The final result is a novel work that expand our kwnoledge on NGS metagenomics applied to particular types of samples, thus providing a base for further studies. For these reasons, I recommend the paper for the publication.

Reviewer 4 Report

The updated manuscript reads much better. Authors have incorporated and addressed the points raised and made the relevant changes where necessary.